# Evaluation for the Genetic Association between Store-Operated Calcium Influx Pathway (STIM1 and ORAI1) and Human Hepatocellular Carcinoma in Patients with Chronic Hepatitis B Infection

**DOI:** 10.3390/biology9110388

**Published:** 2020-11-09

**Authors:** Lalu Muhammad Irham, Wan-Hsuan Chou, Yu-Shiuan Wang, Wirawan Adikusuma, Henry Sung-Ching Wong, Dyah Aryani Perwitasari, Wan-Chen Huang, Ben-Kuen Chen, Hwai-I Yang, Wei-Chiao Chang

**Affiliations:** 1Department of Clinical Pharmacy, School of Pharmacy, College of Pharmacy, Taipei Medical University, Taipei 11031, Taiwan; lalu.irham@pharm.uad.ac.id (L.M.I.); s700081@gmail.com (W.-H.C.); adikusuma28@gmail.com (W.A.); miningyue@gmail.com (H.S.-C.W.); 2Faculty of Pharmacy, University of Ahmad Dahlan, Yogyakarta 55164, Indonesia; diahperwitasari2003@yahoo.com; 3Master Program for Clinical Pharmacogenomics and Pharmacoproteomics, School of Pharmacy, Taipei Medical University, Taipei 11031, Taiwan; 4Institute of Cellular and Organismic Biology, Academia Sinica, Taipei 115, Taiwan; yswang1004@gmail.com (Y.-S.W.); wanchen.huang@gmail.com (W.-C.H.); 5Ph.D Program in Clinical Drug Development of Chinese Herbal Medicine, College of Pharmacy, Taipei Medical University, Taipei 11031, Taiwan; 6Departement of Pharmacy, University of Muhammadiyah Mataram, Mataram 83127, Indonesia; 7Ph.D. Program for translational Medicine, College of Medical Science and Technology, Taipei Medical University, Taipei 11031, Taiwan; 8Department of Pharmacology, College of Medicine, National Cheng Kung University, Tainan 701, Taiwan; 9Genomics Research Center, Academia Sinica, Taipei 115, Taiwan; 10Institute of Clinical Medicine, National Yang-Ming University, Taipei 112, Taiwan; 11Integrative Therapy Center for Gastroenterologic Cancers, Wan Fang Hospital, Taipei Medical University, Taipei 116, Taiwan; 12Department of Medical Research, Shuang Ho Hospital, Taipei Medical University, New Taipei City 23561, Taiwan

**Keywords:** chronic hepatitis B (CHB), hepatocellular carcinoma (HCC), hepatitis B virus (HBV), *STIM1*, *ORAI1*, single-nucleotide polymorphism (SNP)

## Abstract

**Simple Summary:**

In this study, we systematically evaluated the genetic susceptibility between the store-operated calcium (SOC) influx pathway (stromal interaction molecule 1 (STIM1) and ORAI1) and human hepatocellular carcinoma (HCC) in patients with chronic hepatitis B (CHB) infection. In total, 3631 patients with CHB were recruited, forty polymorphisms of *STIM1* and *ORAI1* were comprehensively analyzed. Three single-nucleotide polymorphisms (SNPs) of *STIM1* (rs6578418, rs11030472, and rs7116520) and one SNP of *ORAI1* (rs6486795) showed a trend of being significantly associated with HCC. In particular, our functional studies (images from total internal reflection fluorescence microscopy and transwell migration assay) revealed that calcium (Ca^2+^) signaling is essential for the migration of HCC. Based on such a comprehensively screening in 3631 patients with chronic hepatitis, our results indicate the important role of store-operated calcium pathways in HCC.

**Abstract:**

Hepatocellular carcinoma (HCC) often develops from chronic hepatitis B (CHB) through replication of hepatitis B virus (HBV) infection. Calcium (Ca^2+^) signaling plays an essential role in HBV replication. Store-operated calcium (SOC) channels are a major pathway of Ca^2+^ entry into non-excitable cells such as immune cells and cancer cells. The basic components of SOC signaling include the *STIM1* and *ORAI1* genes. However, the roles of *STIM1* and *ORAI1* in HBV-mediated HCC are still unclear. Thus, long-term follow-up of HBV cohort was carried out in this study. This study recruited 3631 patients with chronic hepatitis (345 patients with HCC, 3286 patients without HCC) in a Taiwanese population. Genetic variants of the *STIM1* and *ORAI1* genes were detected using an Axiom CHB1 genome-wide array. Clinical associations of 40 polymorphisms were analyzed. Three of the *STIM1* single-nucleotide polymorphisms (SNPs) (rs6578418, rs7116520, and rs11030472) and one SNP of *ORAI1* (rs6486795) showed a trend of being associated with HCC disease (*p* < 0.05). However, after correction for multiple testing, none of the SNPs reached a significant level (*q* > 0.05); in contrast, neither *STIM1* nor *ORAI1* showed a significant association with HCC progression in CHB patients. Functional studies by both total internal reflection fluorescence images and transwell migration assay indicated the critical roles of SOC-mediated signaling in HCC migration. In conclusion, we reported a weak correlation between *STIM1/ORAI1* polymorphisms and the risk of HCC progression in CHB patients.

## 1. Introduction

Hepatitis B virus (HBV) infection remains the most common chronic viral infection in the world. Despite the development of a vaccine, HBV remains a serious problem worldwide. The World Health Organization (WHO) reported around 257 million new cases and 887,000 deaths due to HBV infection in 2018 [1]. Among them, most patients died from HBV-related complications, including liver cirrhosis and hepatocellular carcinoma (HCC) [2]. HCC, a primary liver cancer, is the fifth most often diagnosed cancer and the second cause of cancer-related deaths among all cancers worldwide [3]. It was also recorded as the second leading cause of cancer-related deaths in East Asia and sub-Saharan Africa and the sixth most common cancer in western countries [4,5]. The incidence of HCC is heterogeneously distributed geographically, with more than 80% of HCC cases occurring in East Asia and sub-Saharan Africa, and China alone accounting for over 50% of global HCC cases [6].

Many factors may contribute to the development of HCC from HBV infection, including host factors (such as a male gender, an older age, being seropositive of the hepatitis B surface antigen (HBsAg), and genetic variants), environmental factors (such as aflatoxin B1 and alcohol consumption), and viral HBV genotypes [6,7,8,9]. Moreover, host genetic polymorphisms of several genes, as reported in Asian studies, could be associated with clinical outcomes of HBV-related HCC, including kinesin family member 1b (*KIF1B*) rs17401966 [10], major histocompatibility complex, class II, DQ alpha 1 (*HLA-DQA1*) rs9272105, glutamate ionotropic receptor kainate type subunit 1 (*GRIK1*) rs455804 [11], major histocompatibility complex, class II, DQ alpha 2 (*HLA-DQA2*) rs9275319, signal transducer and activator of transcription 4 (*STAT4*) rs7574865 [12], and phosphatidylinositol-4,5-bisphosphate 3-kinase catalytic (*PIK3CA*) rs17849071 [13].

The HBV X protein (HBx), the smallest HBV protein, was reported to mediate the viral replication process in hepatocytes through elevating the intracellular calcium (Ca^2+^) concentration [14]. Elevation of intracellular Ca^2+^ results from Ca^2+^ release from the endoplasmic reticulum (ER), a major intracellular Ca^2+^ store, and Ca^2+^ influx through Ca^2+^ channels [15,16]. There are two main proteins involved in Ca2+ elevation in unexcitable cells: stromal interaction molecule 1 (STIM1) located in the ER and ORAI1 expressed by plasma membranes [17]. STIM1, a protein involved in the store-operated Ca^2+^ (SOC) entry process, was confirmed to be required by cluster of differentiation 4-positive (CD4^+^) and cluster of differentiation 8-positive (CD8+) T-cell antiviral immunity [18]. STIM1 acts as a Ca^2+^ sensor, which is able to interact with ORAI1 to trigger Ca^2+^ influx [17]. Previous studies have reported that defective regulation of SOC signaling contributes to the pathogenesis of several diseases including cancer progression, infections, allergies, and hemostasis [17]. However, the roles of *STIM1* and *ORAI1* in HBV-mediated HCC remain unclear. Herein, we conducted a genetic association study to address this question.

## 2. Materials and Methods

### 2.1. Study Subjects

This study is part of the Risk Evaluation of Viral Load Elevation and Associated Liver Disease/Cancer-Hepatitis B Virus (REVEAL-HBV) study, which is a community-based cohort study in Taiwan of individuals infected with HBV aged 30–65 years. The workflow of the study is depicted in Figure 1. In total, 3631 HBV patients (345 patients with HCC and 3286 patients without HCC) met the inclusion criteria (i.e., seronegative for antibodies against the hepatitis C virus/anti-HCV, adequate serum sample of hepatitis B early antigen (HBeAg), seropositive for the hepatitis B surface antigen (HBsAg), and serum levels of alanine aminotransferase (ALT) and HCC-free patients at study entry) [19]. The period of study recruitment was 1991–1992 with a follow-up until 2014 in seven regions of Taiwan (i.e., Sanchi, Chutung, Potzu, Kaoshu, Makung, Hushi, and Paisha). All of the study participants were ethnic Chinese (i.e., Taiwanese). They provided written informed consent before participation as a declaration, agreeing that we could conduct an interview, collect a blood specimen, and conduct various serologic and biochemical assays. This project was approved by the ethics committees at Academia Sinica (AS-IRB-BM-15017) Taipei, Taiwan. All participants were interviewed in person using a structured questionnaire administered by well-trained public health nurses. Sociodemographic characteristics of each patient were queried, including their medical and surgical histories, and any family history of HCC.

### 2.2. Clinical Evaluation

All participants in the REVEAL-HBV study were followed-up every 6–12 months to check for a clinical evaluation of HBV characteristics including determining the HBsAg level, the serostatus of HBsAg, the serostatus of HBeAg, and the ALT level. The serostatuses of HBsAg, HBeAg, and ALT were tested using commercial kits. Levels of HBsAg and HBeAg were quantified with a radioimmunoassay (Abbott Laboratories, North Chicago, IL, USA). Serum HBsAg levels were quantified using a Roche Elecsys HBsAg II Quant assay. ALT was quantified using a serum chemistry autoanalyzer (model 736; Hitachi, Tokyo, Japan) with commercial reagents. The HBV genotype was determined in those with detectable serum levels of HBV DNA by a melting curve analysis.

### 2.3. Ascertainment of HCC

All participants of this study were ascertained to not have HCC at the time of study entry, and cases of HCC during follow-up were determined by computerized linkage of data with information from the National Cancer Registry in Taiwan. The HCC diagnosis was based on pathological examination of hepatic specimens, positive lesions with confirmation, which included at least two different imaging techniques (angiogram, computed tomography, or abdominal ultrasonography), or positive lesions by one imaging technique accompanied by an α-fetoprotein level of ≥400 ng/mL.

### 2.4. Genotyping of SNPs

Our study is part of the REVEAL-HBV cohort study, a community-based cohort (consisting of 345 patients with HCC during follow-up and 3286 patients without HCC). Human genomic DNA was extracted from peripheral blood leukocytes using Qiagen commercial kits (QIAamp DNA Blood Maxi Kit) and standard methods. Axiom CHB1 genome-wide array was applied to discover genes potentially associated with HBV-related HCC.

### 2.5. SNP Annotation Data Query

In order to evaluate the relationship between SNPs and profiles of gene expression, we confirmed it by examining the expression quantitative trait loci (e-QTL) through Genotype-Tissue Expression (GTEx) portal database (http://www.gtexportal.org/home/), which contains the expressions of genes in a variety of tissues.

### 2.6. Cell Culture

The liver cancer cell lines Huh 7 and HepG2 were cultured in Dulbecco’s modified Eagle medium (DMEM; Invitrogen, 32571036) and incubated at 37 °C with 5% CO_2_ for the complete culture medium, and the DMEM was contained with 100 mg/mL streptomycin, 100 U/mL penicillin (Invitrogen, 15140122) and 10% (vol/vol) bovine calf serum (Invitrogen, 10437-028).

### 2.7. Plasmids Transfection

The plasmids STIM1-yellow fluorescent protein (STIM1-YFP) (Addgene, #19754) and Orai1-monomeric red fluorescent proteins (Orai1-mCherry) were used in real-time total internal reflection fluorescence microscope (TIRF) imaging systems. For expression of the plasmids in cells, the plasmids were transfected with TurboFect (Thermo Fisher Scientific, R0532) for 24 h. The cells were re-seeded on Lab-Tek chambered cover glass (Thermo Fisher Scientific, NUC155383) before observed with a microscope.

### 2.8. Transwell Migration Assay

To examine the effect of SOC inhibitor, 2-aminoethoxydiphenylborate (2-APB) on cell motility, the Huh 7 or HepG2 cells were pre-treated with 2-APB for 30 minutes. The 1 × 10^5^ cells were seeded in the transwell insert (BD, 353097) contained with serum-free medium and the complete medium was added into the 24-well plate as a chemo-attractant. After 24 or 48 hours, the inserts were washed with phosphate-buffered saline (PBS) and the cells in the inserts were removed. The cells attached to the reverse side of the insert were fixed with ice cold methional and stained with crystal violet. The migrated cells were photographed by an inverted microscope (LEICA, DM IL LED) with 20× objective lens and analyzed with Image J software.

### 2.9. Total Internal Reflection Fluorescence Microscopy (TIRF)

In order to observe the effects of 2-APB on thapsigargin (TG)-induced SOC activity, the STIM1-YFP and Orai1-mCherry co-expressed Huh 7 cells were subcultured on Lab-Tek chambered cover glass before imaging was taken by a microscope. The total internal reflection fluorescence microscope (TIRFM) system was built on an inverted microscope (Olympus IX81, Tokyo, Japan) with a high-sensitivity Electron Multiplying Charge Coupled Device (EMCCD) camera (iXOn3897, Andor Technology, Tubney Woods, Abingdon, UK). In order to observe the fluorescent signal on the cell membrane, the UPONAPO 60x OTIRF objective lens (NA: 1.49; Olympus) was applied to achieve TIRF images. The fluorescent proteins labeled with YFP and mCherry were excited with 488 and 532 nm solid lasers, respectively, and the program was driven by Xcellence software (Olympus imaging software).

### 2.10. Statistical Analysis

Distributions of single-nucleotide polymorphisms (SNPs) of *STIM1* and *ORAI1* were tested for Hardy–Weinberg equilibrium (HWE) in order to evaluate allelic distributions between the two populations. All SNPs in our study met the HWE condition (all *p* > 0.05) with minor allele frequency (MAF) of >5%. Linkage disequilibrium (LD) was evaluated with Haploview software version 4.2 (Broad Institute, Cambridge, MA, USA). LD was examined for SNP pairs, and haplotype blocks were defined using the default setting of the Haploview software version 4.2. We used logistic regression analyses to obtain odds ratio (OR) of HCC cases adjusted using age, gender, ALT, HBsAg level, HBeAg serostatus, and viral genotype. Associations between SNPs and HCC under the genotype, dominant, and allelic models were determined using the R “SNPassoc” package analyses in the R environment (https://cran.r-project.org/ and https://www.r-project.org/), multiple testing correction was performed using the false discovery rate (FDR), and q-values of <0.05 were determined to indicate statistical significance. Haplotype associations with HCC were determined with Haploview software version 4.2.

## 3. Results

### 3.1. Basic Characteristics of Chronic Hepatitis B-Infected Patients

We analyzed the data from 3631 chronic HBV-infected subjects, which included 345 chronic hepatitis B (CHB) patients with HCC progression and 3286 CHB patients without HCC progression. ALT, the HBsAg levels, and the HBeAg serostatus were monitored, and the mean and range of age, gender, and viral genotype are shown in Table 1. Results indicated that the average age of CHB patients was 45.7 years, and male gender was more dominant (2196 subjects) compared to female gender (1435 subjects). Serological marker of ALT level indicated that patients with an ALT level of <45 U/L were more dominant (3405 subjects) compared to patients with an ALT level of >45 U/L (226 subjects). In case of the serological marker of HBsAg level, there are 1519 subjects with a HBsAg level >1000 IU/mL, 946 subjects with a HBsAg level <1000 IU/mL and 918 subjects with a HBsAg level <100 IU/mL. Multivariate analyses showed that participants of an older age had a significantly higher risk of suffering from HCC, with an OR of 2.32 for those 40–49 years, an OR of 3.47 for those 50–59 years, and an OR of 3.8 for those >60 years after adjusting for gender and serum ALT. Male patients had a 2.53-fold tendency of a higher risk compared to female CHB patients after adjusting for age and serum ALT. Patients with an ALT level of >45 U/L had a 3.7-fold tendency of a higher risk compared to CHB patients with an ALT level of <45 U/L after adjusting for age and gender.

Patients with a HBsAg level of >1000 IU/mL had a 5.24-fold higher risk of suffering from HCC compared to those with a HBsAg level of <1000 IU/mL after adjusting for age, ALT, and gender. Subjects who were HBeAg positive also had a 7.87-fold higher risk of HCC compared to those who were negative for HBeAg after adjusting for age, ALT, and gender. Subjects with the viral genotype type C had a 3.06-fold higher risk of suffering from HCC compared to those with genotype B or B+C after adjusting for age, ALT, and gender (Table 1).

### 3.2. STIM1 and ORAI1 SNPs with Minor Allelic Frequencies in Different Populations

Table 2 shows the MAFs of *STIM1* and *ORAI1* polymorphisms in different populations worldwide (e.g., Asian, African, American, European, and Taiwanese). MAFs of African, American, European, and Asian populations were extracted from the HaploReg browser version 4.1 (https://pubs.broadinstitute.org/mammals/haploreg/haploreg.php), and MAFs of the Taiwan Biobank (TWB) were obtained from the Taiwan View website (https://taiwanview.twbiobank.org.tw/index). In addition, allele frequencies in the current study are similar to those in reference to Asians, including the TWB. All SNPs met the HWE condition with an MAF of >5%.

### 3.3. Genetic Association of STIM1 and ORAI1 with HCC

Our study assessed 37 *STIM1* polymorphisms and three *ORAI1* polymorphisms (Table 3). Three genetic models (allelic, genotype, and dominant) were applied to test the association between HCC and these SNPs. rs6578418, rs11030472, and rs7116520 of *STIM1* and rs6486795 of *ORAI1* showed a trend of being significantly associated with HCC disease (*p* < 0.05). However, after correction for multiple testing, none of the SNPs reached a significant level (*q* > 0.05) (Table 3).

For *STIM1*, rs6578418 showed a weak association under the allelic model (*p* = 0.048) and dominant model (*p* = 0.044). In addition, rs11030472 showed a weak association under the dominant model (*p* = 0.0296), and rs7116520 showed a weak association with HCC under the genotypic (*p* = 0.0330), dominant (*p* = 0.0093), and allelic models (*p* = 0.0165). In *ORAI1*, only rs6486795 among the three SNPs that we evaluated showed a trend of being associated with HCC after adjusting for serum ALT, gender, and age. rs6486795 showed a weak association under the allelic model, with *p* = 0.0135. However, after correction for multiple testing, none of the SNPs reached a significant level (Table 3).

### 3.4. Haplotype Analysis of STIM1 and ORAI1

We further performed haplotype analysis. Results showed weak correlations of *STIM1* (Appendix A) and *ORAI1* (Appendix A) haplotypes with HCC disease. For *STIM1*, both GGTAGCAT haplotype of block six and AGCGG haplotype of block seven had trends of being associated with HCC at *p* = 0.0163 and *p* = 0.015, respectively (Appendix A). In addition, results for *ORAI1* indicated a trend of HBV carriers with the TTC (*p* = 0.013) haplotype being more susceptible to associate with HCC progression than those with the CTT, CCC, or CTC haplotype (Appendix A). Unfortunately, none of these haplotypes of *STIM1* and *ORAI1* were still significant after applying the multiple correction.

### 3.5. Annotation of Expression Quantitative Trait Loci (e-QTLs) for STIM1 and ORAI1

To elaborate the relationship between the genetic polymorphisms and gene expression, the publicly available database GTEx portal (http://www.gtexportal.org/home/) was utilized to study the tissue expression quantitative trait loci. For rs6578418 and rs11030472, we could not find the gene expression data on the GTEx portal database due to limited available tissues (Appendix A). While the trend of *STIM1* rs7116520 showed that GG genotype had a higher *STIM1* expression compared to those with AG and AA genotypes (Appendix A). Furthermore, *ORAI1* rs6486795 with CC genotype was associated with higher *ORAI1* expression compared to those with TC and TT genotypes in a variety of tissues (e.g., whole blood, esophagus, thyroid, heart, stomach, pancreas, muscle, colon).

### 3.6. 2-APB Inhibited Liver Cancer Cell Migration via Suppressed STIM1-ORAI1 Colocolization

STIM1 and ORAI1 are two basic components of SOC as a major pathway of calcium entry in non-excitable cells, especially in the cancer cells [17]. To validate the roles of SOC influx pathway (STIM1 and ORAI1) in HCC cell progression, 2-APB, a SOC inhibitor was applied. As shown in Figure 2A, the colocalization of STIM1 and ORAI1 was inhibited by 2-APB pre-treatment for 30 minutes in Huh 7 cells. We further analyzed the inhibitory effects of 2-APB in cell migration by using Huh 7 and HepG2 cell lines. Results showed that liver cancer cells migration (Huh 7 and HepG2) significantly decreased by 2-APB pre-treatment compared to the control group (Figure 2B,C).

## 4. Discussion

Hepatocellular carcinoma (HCC) is the second highest cause of cancer-related deaths worldwide [3]. Despite the fact that CHB is a major cause of HCC [2], only a fraction of CHB-infected patients develops HCC during their lifetime. Therefore, it is important to determine the potential protective or risk factors related to HCC susceptibility. Our investigation comprised a cohort study with 3631 CHB patients consisting of 345 CHB patients with HCC progression and 3286 CHB patients without HCC progression. This study was part of the REVEAL-HBV cohort study, which previously revealed that male patients, older patients and those with a higher ALT level, with HBeAg positive or who had viral genotype C had a significantly higher risk of CHB-related HCC progression [8,19,20]. Calcium signaling has been shown as a critical factor for viral replication during HBV pathogenesis [14]. Mitochondrial Ca^2+^ uptake, SOC pathway and the C-terminal of ORAI1 protein involved in the regulatory mechanism of HBX protein-modulated cytosolic calcium increase [15,21,22]. In this study, we systematically evaluated the genetic susceptibility between the SOC influx pathway (STIM1 and ORAI1) and human HCC in patients with CHB infection. In total, 3631 patients with chronic hepatitis were recruited. Forty polymorphisms in the SOC pathway were comprehensively analyzed. Three SNPs of *STIM1* (rs6578418, rs11030472, and rs7116520) and one SNP of *ORAI1* (rs6486795) showed a trend of correlation with HCC. However, none of these SNPs reached the significance after multiple correction. Interestingly, the trend of risk alleles of four variants (rs6578418, rs11030472, rs7116520, and rs6486795) showed higher frequencies in Asian populations compared to European and American populations. The trend of the G allele as a risk allele of rs11030472 represented 16–19% frequencies, in current study (19%), in the TWB (18%) and in the Asian region (16%), compared to European, African, and American at 0%. In addition, variant rs6486795 of the C allele showed an allelic frequency in an Asian population of around 38%, and our study also showed a 37% frequency compared to American and European populations at 28% and 21%, respectively. For two variants, the G allele of rs6578418 and rs7116520 exhibited homogeneity in Asian (rs6578418, 8% and rs7116520, 47%), European (rs6578418, 8% and rs7116520, 45%) and American (rs6578418, 9% and rs7116520, 46%) populations, respectively. It is likely that variations in allelic frequencies among different populations may play a role in the observed different results in other populations. However, we cannot rule out the possibility that other low/high frequency genetic polymorphisms of *STIM1* and *ORAI1* may contribute to HCC progression in CHB.

The roles of the *STIM1* and *ORAI1* genes were also determined in other diseases. For example, rs6486795 of the *ORAI1* gene was associated with various diseases, including atopic dermatitis in Japanese and Taiwanese populations [23] and nephrolithiasis in a Taiwanese population [24]. In addition to these results, other studies also reported that polymorphisms of *ORAI1* are involved in other diseases such as ankylosing spondylitis and Kawasaki disease [25,26]. Results of haplotype analyses indicated that HBV carriers with the GGTAGCAT haplotype had a trend of being associated with a HCC risk; additionally, those with the AGCGG haplotype also showed a trend of being more susceptible to HCC progression. Results for *ORAI1* also indicated a trend of HBV carriers with the TTC haplotype being more susceptible to being associated with a HCC risk than those with the CTT, CCC, or CTC haplotype. However, we found no significant association after applying the multiple correction.

Calcium influx through the SOC channel plays a pivotal role in regulating many signaling pathways related to cancer progression, infections, allergies, and hemostasis [17], such as colorectal cancer [27,28], cervical cancer [29] melanomas [30], and lung cancer [31] and breast cancer [32]. Functional studies have indicated the inhibition of *STIM1* and *ORAI1* in the metastasis of cancer cells [32]. Mutations of *STIM1* and *ORAI1* were also reported to affect patients with viral, bacterial, and fungal infections via abolishing the SOC influx and immune system [33,34]. Thus, the correlations between genetic factors and HCC progression in HBV infection might due to the increase in Ca^2+^ levels that initially commence from SOC influx, which is the predominant mechanism of Ca^2+^ entry in hepatocytes [21]. The HBx protein is the smallest HBV protein that has been reported in mediating HBV replication and development of HCC through elevation of cytosolic calcium signals [21,35]. Therefore, an association study of *STIM1* and *ORAI1* polymorphisms for HBV DNA level was also examined in this study; however, the results did not show a significant association (data not shown). Elevation of Ca^2+^ through the SOC channel has an important role in hepatocytes. Indeed, the elevation of cytosolic calcium involves the replication of the HBV and hepatitis C virus (HCV), which might cause HCC [15,36]. Here, our study also indicated that knockdown of the SOC pathway by 2-APB prevented cell migration in two liver cancer cell lines (Huh 7 and HepG2). We acknowledge that the weak correlation between *STIM1* and *ORAI1* polymorphisms and the risk of HCC progression in CHB patients may be due to the modest sample size (3631 CHB patients), which led to the reduced power of the statistical analysis. Another possibility is that the majority of polymorphisms were located in the non-coding region. The genetic effects of each polymorphism for the expression of the store-operated calcium channel are mild.

## 5. Conclusions

Our study indicated that inhibition of the SOC pathway is able to block cell migration in two liver cancer cell lines (Huh 7 and HepG2). Although genetic polymorphisms of the SOC pathway (*STIM1* and *ORAI1*) are not significantly associated with HCC progression after multiple correction, three SNPs of *STIM1* (rs6578418, rs11030472, and rs7116520) and one SNP of *ORAI1* (rs6486795) showed a borderline significant trend that should be particularly focused on in future studies.

## Figures and Tables

**Figure 1 biology-09-00388-f001:**
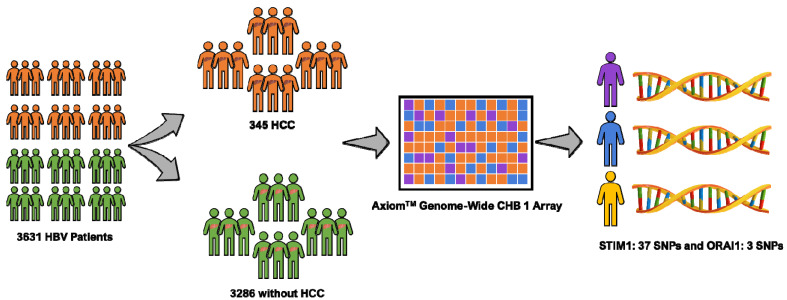
Data from 3631 hepatitis B virus (HBV) patients including 3286 subjects without hepatocellular carcinoma (HCC) and 345 subjects with HCC were analyzed. Axiom-chronic hepatitis B -(CHB) genome-wide array was used for genotyping. A total of 40 single nucleotide polymorphism (SNPs) were assessed (37 *STIM1* and three *ORAI1*) through this method.

**Figure 2 biology-09-00388-f002:**
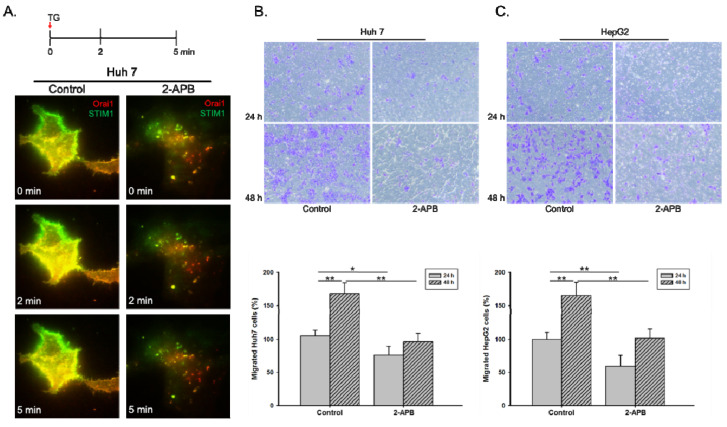
Inhibition of STIM1-ORAI1 colocalization by 2-APB pre-treatment leads to reduced liver cancer cell migration ability. The Huh 7 cells were pre-treated with store-operated calcium (SOC) inhibitor, 2-APB, for 30 min. (**A**). The cells were co-transfected with STIM1-YFP and Orai1-mCherry and reseeded on Lab-Tek chambered cover glass for 24 h. The inhibition effect of 2-APB on thapsigargin (TG)-induced SOC activity was observed with time-lapse TIRFM images. The inhibition effect of 2-APB on (**B**) Huh 7 or (**C**) HepG2 cell migration ability was examined by transwell migration assay. Statistically significant data are indicated by * for *p* < 0.05 and ** for *p* < 0.01.

**Table 1 biology-09-00388-t001:** Baseline characteristics and multivariate adjusted odd ratios (ORs) of hepatocellular carcinoma (HCC) according to various risk factors.

Characteristic	Overall *N* = 3631 (%)	Without HCC (*N* = 3286)	With HCC (*N* = 345)	OR (95% CI)	*p* Value
Age, mean years (SD)	45.77	45.32 (9.65)	50.08 (8.87)	-	
Age range (years)	30~65	30~65	30~65	-	
30~39	1194 (32.88)	1140 (34.69)	54 (15.65)	Ref	
40~49	1031 (28.39)	931 (28.33)	100 (28.98)	2.32 (1.64~3.29)	**<0.001 ***
50~59	1049 (28.90)	911 (27.72)	138 (40.00)	3.47 (2.48~4.84)	**<0.001 ***
≥60	357 (9.83)	304 (9.25)	53 (15.36)	3.8 (2.53~5.72)	**<0.001 ***
Gender					
Female, *n* (%)	1435 (39.52)	1365 (41.54)	70 (20.29)	Ref	
Male, *n* (%)	2196 (60.48)	1921 (58.46)	275 (79.71)	2.53 (1.86~3.45)	**<0.001 ***
ALT (U/L)					
<45	3405 (93.78)	3118 (94.89)	287 (83.18)	Ref	
≥45	226 (6.22)	168 (5.12)	58 (16.82)	3.7 (2.65~5.17)	**<0.001 ***
HBsAg level (IU/mL)£					
<100	918 (27.14)	880 (28.59)	38 (12.46)	Ref	
≤1000	946 (27.96)	883 (28.69)	63 (20.66)	2.01 (1.32~3.07)	**0.001 ***
≥1000	1519 (44.90)	1315 (42.72)	204 (66.89)	5.24 (3.6~7.63)	**<0.001 ***
HBeAg serostatus †					
Negative	2962 (84.70)	2765 (87.58)	197 (57.94)	Ref	
Positive	535 (15.30)	392 (12.42)	143 (42.06)	7.87 (5.95~10.42)	**<0.001 ***
Viral genotype ††					
B+(BC)	1734 (65.28)	1607 (68.04)	127 (43.19)	Ref	
C	922 (34.71)	755 (31.96)	167 (56.80)	3.06 (2.36,3.97)	**<0.001 ***

SD, standard deviation; ALT, alanine aminotransferase; HBsAg, hepatitis B surface antigen; HBeAg, hepatitis B early antigen. ***** Significant *p* value (<0.05) are in **bold**. £ Data were only available for 3383 samples. † Data were only available among the Risk Evaluation of Viral Load Elevation and Associated Liver Disease/Cancer-Hepatitis B Virus (REVEAL-HBV) cohort for 3497 samples. †† Data were available for 2656 individuals with detectable HBV DNA levels (≥300 copies/mL).

**Table 2 biology-09-00388-t002:** Minor allelic frequencies of single-nucleotide polymorphisms (SNPs) in this study.

SNP	Position (hg38) (bp)	Gene	Allele		**Minor Allelic Frequencies** **(MAFs)**						
			Major	Minor	AFR	AMR	ASN	EUR	TWB	Ours	HWE
rs4243966	Chr11:3864993	*STIM1*	T	C	0.05	0.07	0.04	0.06	0.04	0.04	0.95
rs7943201	Chr11:3865531	*STIM1*	G	A	0.33	0.45	0.36	0.37	0.38	0.37	0.62
rs11030122	Chr11:3865946	*STIM1*	C	G	0.09	0.23	0.34	0.33	0.34	0.34	0.65
rs7951076	Chr11:3866612	*STIM1*	G	A	0.11	0.31	0.28	0.29	0.25	0.26	0.74
rs7952083	Chr11:3869730	*STIM1*	A	C	0.19	0.41	0.38	0.49	0.38	0.39	0.60
rs7120828	Chr11:3879685	*STIM1*	C	T	0.48	0.26	0.34	0.22	0.35	0.34	0.66
rs10458894	Chr11:3892042	*STIM1*	C	T	0.23	0.39	0.31	0.35	0.30	0.31	0.68
rs7120683	Chr11:3894732	*STIM1*	T	G	0.43	0.27	0.34	0.22	0.35	0.34	0.65
rs10835262	Chr11:3895276	*STIM1*	G	A	0.19	0.23	0.34	0.34	0.34	0.34	0.66
rs4622250	Chr11:3897371	*STIM1*	C	T	0.09	0.22	0.30	0.33	0.25	0.25	0.75
rs75197750	Chr11:3901583	*STIM1*	C	T	0.03	0.04	0.05	0.03	0.03	0.05	0.98
rs10500589	Chr11:3902756	*STIM1*	T	C	0.02	0.23	0.01	0.02	0.21	0.22	0.77
rs11030209	Chr11:3903119	*STIM1*	A	C	0.21	0.23	0.29	0.33	0.25	0.25	0.74
rs10835270	Chr11:3903372	*STIM1*	G	T	0.31	0.28	0.31	0.38	0.27	0.27	0.72
rs11030210	Chr11:3903482	*STIM1*	G	T	0.01	0.07	0.03	0.06	0.04	0.04	0.95
rs7929653	Chr11:3904546	*STIM1*	G	A	0.31	0.46	0.40	0.43	0.43	0.44	0.56
rs6578418	Chr11:3905002	*STIM1*	C	G	0.08	0.09	0.08	0.08	0.08	0.07	0.93
rs10835272	Chr11:3905008	*STIM1*	G	A	0.43	0.26	0.47	0.14	0.46	0.45	0.55
rs10742189	Chr11:3907522	*STIM1*	A	C	0.33	0.37	0.43	0.49	0.39	0.39	0.60
rs7129444	Chr11:3911802	*STIM1*	C	T	0.36	0.29	0.35	0.42	0.31	0.32	0.67
rs7118422	Chr11:3912065	*STIM1*	C	T	0.22	0.48	0.48	0.48	0.46	0.45	0.55
rs4910863	Chr11:3917892	*STIM1*	T	C	0.08	0.46	0.46	0.47	0.44	0.43	0.56
rs11030264	Chr11:3919001	*STIM1*	A	G	0.04	0.49	0.43	0.46	0.41	0.41	0.58
rs7924984	Chr11:3953358	*STIM1*	G	A	0.30	0.33	0.46	0.35	0.47	0.47	0.52
rs2412338	Chr11:3956291	*STIM1*	G	T	0.31	0.33	0.46	0.35	0.47	0.48	0.51
rs10835402	Chr11:3969684	*STIM1*	T	C	0.02	0.24	0.30	0.32	0.28	0.28	0.72
rs11030472	Chr11:3978105	*STIM1*	A	G	0.00	0.00	0.16	0.00	0.18	0.19	0.80
rs11030478	Chr11:3979345	*STIM1*	G	A	0.11	0.08	0.15	0.02	0.20	0.20	0.79
rs11030486	Chr11:3980034	*STIM1*	C	T	0.11	0.08	0.15	0.02	0.20	0.20	0.79
rs12284835	Chr11:3997798	*STIM1*	G	A	0.08	0.03	0.05	0.08	0.04	0.05	0.95
rs727152	Chr11:3998535	*STIM1*	A	G	0.50	0.12	0.10	0.16	0.10	0.10	0.89
rs2959081	Chr11:4005810	*STIM1*	T	C	0.12	0.48	0.44	0.49	0.41	0.40	0.60
rs11030639	Chr11:4017826	*STIM1*	A	G	0.30	0.26	0.32	0.33	0.31	0.31	0.68
rs7116520	Chr11:4032147	*STIM1*	A	G	0.10	0.46	0.47	0.45	0.42	0.42	0.58
rs1442725	Chr11:4068357	*STIM1*	C	T	0.48	0.10	0.05	0.08	0.05	0.05	0.94
rs11030841	Chr11:4077783	*STIM1*	G	A	0.16	0.22	0.21	0.12	0.26	0.26	0.73
rs4910882	Chr11:4081185	*STIM1*	G	A	0.16	0.22	0.21	0.12	0.26	0.26	0.73
rs6486795	Chr12:121638011	*ORAI1*	T	C	0.36	0.28	0.38	0.21	0.38	0.37	0.62
rs74936888	Chr12:121638300	*ORAI1*	T	C	0.07	0.00	0.13	0.01	0.08	0.08	0.92
rs3741595	Chr12:121641283	*ORAI1*	T	C	0.07	0.24	0.21	0.15	0.26	0.26	0.73

AFR, African; AMR, American; ASN, Asian; EUR, European; TWB, Taiwan Biobank; HWE, Hardy–Weinberg equilibrium. Minor allele frequencies (MAFs) of AFR, AMR, ASN and EUR were extracted from the HaploReg browser version 4.1 (https://pubs.broadinstitute.org/mammals/haploreg/haploreg.php) and MAFs of the TWB were obtained from the Taiwan View website (https://taiwanview.twbiobank.org.tw/index).

**Table 3 biology-09-00388-t003:** Association of *STIM1* and *ORAI1* genes with hepatocellular carcinoma in hepatitis B virus (HBV) patients under allelic, genotypic, and dominant models.

SNP	Gene	Effect Allele	(Allelic) *p* Value ^a^ [*q* Value]	(Genotype) *p* Value ^b^ [*q* Value]	(Dominant) *p* Value ^b^ [*q* Value]
rs4243966	*STIM1*	C	0.790 [0.922]	0.602 [0.900]	0.630 [0.968]
rs7943201	*STIM1*	A	0.922 [0.922]	0.881 [0.943]	0.901 [0.989]
rs11030122	*STIM1*	G	0.580 [0.922]	0.804 [0.900]	0.970 [0.989]
rs7951076	*STIM1*	A	0.380 [0.922]	0.811 [0.900]	0.527 [0.867]
rs7952083	*STIM1*	C	0.449 [0.922]	0.581 [0.900]	0.865 [0.989]
rs7120828	*STIM1*	T	0.889 [0.922]	0.722 [0.900]	0.704 [0.989]
rs10458894	*STIM1*	T	0.382 [0.922]	0.629 [0.900]	0.380 [0.867]
rs7120683	*STIM1*	G	0.504 [0.922]	0.760 [0.900]	0.470 [0.867]
rs10835262	*STIM1*	A	0.788 [0.922]	0.972 [0.972]	0.989 [0.989]
rs4622250	*STIM1*	T	0.655 [0.922]	0.626 [0.900]	0.876 [0.989]
rs75197750	*STIM1*	T	0.354 [0.922]	0.383 [0.900]	0.321 [0.867]
rs10500589	*STIM1*	C	0.428 [0.922]	0.707 [0.900]	0.405 [0.867]
rs11030209	*STIM1*	C	0.651 [0.922]	0.765 [0.900]	0.823 [0.989]
rs10835270	*STIM1*	T	0.274 [0.922]	0.438 [0.900]	0.199 [0.858]
rs11030210	*STIM1*	T	0.860 [0.922]	0.353 [0.900]	0.781 [0.989]
rs7929653	*STIM1*	A	0.656 [0.922]	0.622 [0.900]	0.980 [0.989]
rs6578418	*STIM1*	G	**0.048** [0.922]	0.133 [0.900]	**0.044** [0.527]
rs10835272	*STIM1*	A	0.351 [0.922]	0.241 [0.900]	0.978 [0.989]
rs10742189	*STIM1*	C	0.351 [0.922]	0.241 [0.900]	0.978 [0.989]
rs7129444	*STIM1*	C	0.588 [0.922]	0.816 [0.900]	0.544 [0.867]
rs7118422	*STIM1*	T	0.542 [0.9222]	0.899 [0.943]	0.734 [0.989]
rs4910863	*STIM1*	C	0.634 [0.922]	0.735 [0.900]	0.486 [0.867]
rs11030264	*STIM1*	G	0.697 [0.922]	0.954 [0.972]	0.788 [0.989]
rs7924984	*STIM1*	A	0.450 [0.922]	0.530 [0.900]	0.284 [0.867]
rs2412338	*STIM1*	T	0.382 [0.922]	0.568 [0.900]	0.293 [0.867]
rs10835402	*STIM1*	C	0.292 [0.922]	0.487 [0.900]	0.370 [0.867]
rs11030472	*STIM1*	G	0.074 [0.922]	0.068 [0.900]	**0.029** [0.527]
rs11030478	*STIM1*	A	0.810 [0.922]	0.689 [0.900]	0.859 [0.989]
rs11030486	*STIM1*	T	0.765 [0.922]	0.698 [0.900]	0.806 [0.989]
rs12284835	*STIM1*	A	0.129 [0.922]	0.118 [0.900]	0.110 [0.594]
rs727152	*STIM1*	G	0.339 [0.922]	0.671 [0.900]	0.402 [0.867]
rs2959081	*STIM1*	C	0.138 [0.922]	0.140 [0.900]	0.061 [0.527]
rs11030639	*STIM1*	G	0.132 [0.922]	0.181 [0.900]	0.317 [0.867]
rs7116520	*STIM1*	G	**0.016** [0.922]	**0.033** [0.900]	**0.009** [0.399]
rs1442725	*STIM1*	T	0.204 [0.922]	0.287 [0.900]	0.259 [0.867]
rs11030841	*STIM1*	A	0.140 [0.922]	0.367 [0.900]	0.484 [0.867]
rs4910882	*STIM1*	A	0.156 [0.922]	0.392 [0.900]	0.532 [0.867]
rs6486795	*ORAI1*	C	**0.013** [0.594]	0.129 [0.900]	0.055 [0.527]
rs74936888	*ORAI1*	C	0.050 [0.922]	0.207 [0.900]	0.091 [0.594]
rs3741595	*ORAI1*	C	0.074 [0.922]	0.294 [0.900]	0.195 [0.858]

^a^ Fisher’s exact *p* value. ^b^ Adjusted by alanine transaminase, gender, and age. Significant *p* values are in **bold**. The *q* value is the false discovery rate (FDR) estimation for multi-testing.

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
