# Peer review of "Evaluation for the Genetic Association between Store-Operated Calcium Influx Pathway (STIM1 and ORAI1) and Human Hepatocellular Carcinoma in Patients with Chronic Hepatitis B Infection"

_biology, 2020, doi:10.3390/biology9110388_

Round 1
Reviewer 1 Report
The manuscript improved significantly compared to the last version. The addition of cell culture experiments supplements the sequencing results nicely. I therefore agree to publish the manuscript.
Reviewer 2 Report
Authors have addressed all questions and concerns.
This manuscript is a resubmission of an earlier submission. The following is a list of the peer review reports and author responses from that submission.
Round 1
Reviewer 1 Report
Irham et al conducted large-scale profiling for STIM1 and ORAI1 SNP in large HBV patient sets. Though the statistical significance are low for the SNP probabaly due to not enough HCC patients, the data are still quite valuable to the community and have potential clinical benefits. I would happily accept the paper if the author can address my following questions.
- I am quite interested in whether there are gender differences in SNP profile. Namely, since you have a large dataset, could you evaluate whether there is any SNP signature that is more dominant in males or females? And after the gender grouping, if there are any SNPs that showed significant association?
- For all the SNPs you identified, could you please show some examples that some of the SNPs can cause functional alterations to the proteins (mutations in critical domains that may cause a structural change of the protein), especially in the SNPs that shows trends to be associated with the HCC?
Reviewer 2 Report
The manuscript ‘Evaluation of genetic susceptibility between store-operated calcium influx pathway (STIM1 and ORAI1) and human hepatocellular carcinoma in patients with chronic hepatitis B infection’ aims to identify polymorphisms associated with human hepatocellular carcinoma in 3631 Taiwanese patients. Overall, the manuscript presents data that are important to the field, yet a few points need to be clarified. The specific points that resulted in this conclusion are listed below.
The title is not reflective of the research presented in the manuscript. Genetic susceptibility is used in the title however the research is investigating association of SNP with human hepatocellular carcinoma not genetic susceptibility. Genetic susceptibility is also known as genetic predisposition, which cannot be properly evaluated in the current study. Title needs to be revised to reflect the presented study.
Line 28: Remove ‘of’ from ‘in total of 3631’.
Line 32-33. The authors state that ‘our study revealed that calcium signaling is essential for hepatitis B virus replication’. This is not a valid conclusion of the study presented. While there is published research supporting this conclusion, the current study does not. The current study only evaluates SNP in STIM1 and ORAI1 and calcium signaling is not measured. Remove sentence from paragraph.
Line 130: Specify which Qiagen kit was used.
Line 279-281: The authors state that ‘We acknowledge a lack of calcium concentration data as a limitation of this study, which prevented us from understanding its correlation with HCC progression in CHB patients, even though a previous study revealed that the lowest Ca2+ level could be correlated with a decrease in HBV replication’. This is beyond the scope of the current study as the presented research does not evaluate calcium signaling. This sentence and similar statements in the discussion need to be removed.
Font size varies throughout the entire manuscript. Keep font size consistent.
Table 2: Font of column titles is not consistent (minor allelic frequencies). Keep font consistent.
Line 137: Change ‘an’ to ‘a’.
Line 139: Define OR. OR is not defined until line 162.
Line 233: Remove ‘of’ from ‘in total of 3631’.